More coral, more fish? Contrasting snapshots from a remote Pacific atoll

Beldade Ricardo 1 2 3 rbeldade@gmail.com
Mills Suzanne C. 1 2
Claudet Joachim 1 2
Côté Isabelle M. 4
1 CRIOBE, USR 3278 CNRS-EPHE-UPVD , Perpignan Cedex , France
2 Laboratoire d’Excellence “CORAIL” , France
3 MARE—Marine and Environmental Sciences Centre, Faculdade de Ciências da Universidade de Lisboa , Campo Grande, Lisboa , Portugal
4 Earth to Ocean Group, Department of Biological Sciences, Simon Fraser University , Burnaby, BC , Canada
Keith Sally
Electronic publication date: 2015 Jan 29
Publication date: 2015
Volume: 3
Electronic Location ID: e745
Received 2014 Nov 14; Accepted 2015 Jan 7
Copyright: © 2015 Beldade et al.
Copyright year: 2015
Copyright holder: Beldade et al.
License: This is an open access article distributed under the terms of the Creative Commons Attribution License, which permits unrestricted use, distribution, reproduction and adaptation in any medium and for any purpose provided that it is properly attributed. For attribution, the original author(s), title, publication source (PeerJ) and either DOI or URL of the article must be cited.
License URL: https://creativecommons.org/licenses/by/4.0/

Keywords: Coral cover, Coral-fish relationship, Fish assemblages, Fish community structure, Functional relationship, Resilience

Funding: Agence Nationale de Recherche, ANR-11-JSV7-012-01 “Live and Let Die”, ANR-14-CE03-0001 “ACRoSS” LABEX CORAIL “Where do we go now? Fundação para a Ciência e Tecnologia SFRH/BPD/26901/2006 National Sciences and Engineering Council of Canada Funding was provided to SC Mills and J Claudet by the Agence Nationale de Recherche, ANR-11-JSV7-012-01 “Live and Let Die”, ANR-14-CE03-0001 “ACRoSS” and the LABEX CORAIL “Where do we go now?” R Beldade was funded by Fundação para a Ciência e Tecnologia (SFRH/BPD/26901/2006) and IM Côté by a Discovery grant from the National Sciences and Engineering Council of Canada. The funders had no role in study design, data collection and analysis, decision to publish, or preparation of the manuscript.

==============================
Coral reefs are in decline across the globe as a result of overexploitation, pollution, disease and, more recently, climate change. The impacts of changes in coral cover on associated fish communities can be difficult to predict because of the uneven dependence of reef fish species on corals for food, shelter or the three-dimensional structure they provide. We compared live coral cover, reef fish community metrics, and their associations in two surveys of the lagoon of the remote atoll of Mataiva (French Polynesia) carried out 31 years apart. In contrast to the general pattern of decreasing coral cover reported for many parts of the Indo-Pacific region, live coral cover increased 6–7 fold at Mataiva between 1981 and 2012, and fish density nearly doubled. The stable overall reef fish species richness belied a significant shift in community structure. There was little overlap in community composition across years, and fish assemblages in 2012 were more homogeneous in composition than they were in 1981. Changes in species abundance were not clearly related to species-specific reliance on corals. The strong positive relationships between live coral cover and fish diversity and abundance noted in 1981, when coral cover rarely exceeded 10%, were no longer present in 2012, when coral cover rarely fell below this value. The most parsimonious explanation for these contrasting relationships is that, over the combined range of coral cover observed in the 1981 and 2012 snapshots, there is a rapidly asymptotic relationship between coral and fish. Our results, and other data from the south and west Pacific, suggest that fish diversity and abundance might accumulate rapidly up to a threshold of approximately 10% live coral cover. Such a relationship would have implications for our expectations of resistance and recovery of reef fish communities facing an increasingly severe regime of coral reef disturbances.

Introduction

Coral reefs are in decline across the globe as a result of overexploitation, pollution, disease and, more recently, climate change (Hughes et al., 2003; Pandolfi et al., 2003; Bellwood et al., 2004; Eakin et al., 2010). The patterns of loss in coral cover are well established for some reef-bearing regions, such as the Caribbean (Gardner et al., 2003; Schutte, Selig & Bruno, 2010) and the Great Barrier Reef (Ninio et al., 2000; Sweatman et al., 2001). In contrast, for most parts of the Indo-Pacific region, which holds some three-quarters of the world’s coral reefs (Bruno & Selig, 2007), the scale of coral loss was poorly documented until recently. However, a large-scale, long-term quantitative analysis of coral data from this region revealed an early onset of coral decline, generally low coral cover across all sub-regions, and rates of coral loss of 1%–2% annually, even on some of the region’s most intensively managed reefs (Bruno & Selig, 2007). This analysis also highlighted the paucity of long-term data from South Pacific coral reefs, and in particular from the French Polynesian archipelagos (Bruno & Selig, 2007; Chin et al., 2011). The ongoing coral reef crisis threatens reef-building corals, more than one-third of which are at an elevated risk of extinction (Carpenter et al., 2008), and the rich assemblages of other reef-associated species (Bellwood et al., 2004; Pratchett et al., 2008).

The impact of the loss of coral cover on fish diversity and abundance should depend on the extent to which fish rely on live coral (e.g., Graham et al., 2009). Declines in fish species number and density have been observed after major disturbances, such as mass coral bleaching events and population explosions of crown-of-thorns seastars (Acanthaster planci), which have led to extensive coral mortality (e.g., Jones et al., 2004; Feary et al., 2007; Holbrook, Schmitt & Brooks, 2008; Adjeroud et al., 2009; Leray et al., 2012). However, knock-on effects on fish often manifest themselves after a considerable time lag (10–15 years; Garpe et al., 2006; Graham et al., 2007; Alvarez-Filip et al., 2011), suggesting that coral structure may sometimes be as important as live coral cover for the maintenance of diversity in reef fish assemblages (Garpe et al., 2006; Pratchett et al., 2008; Graham & Nash, 2013). Moreover, the repercussions of coral loss are usually uneven among taxa, with species that feed on coral (e.g., butterflyfishes) or depend on live coral for shelter (e.g., small damselfishes) showing more marked declines than less dependent species (Garpe et al., 2006; Graham et al., 2006; Wilson et al., 2006; Wilson et al., 2008; Emslie, Pratchett & Cheal, 2011; Graham & Nash, 2013). Some fish species may even benefit from live coral reduction in the short term, when shifts in benthic composition lead to increased resource availability (Garpe et al., 2006; Graham et al., 2007).

The taxonomically variable reliance of reef fish on live coral suggests that the effect of changes in coral cover on reef fish diversity may be difficult to predict. Although a positive association between live coral cover and fish species richness and or abundance may be expected (e.g., Bell & Galzin, 1984), it is not clear whether this relationship should be constant over all possible coral cover values. Arguably, as coral cover increases, the number of new coral-dependent species or individuals that can be added to existing fish communities should increase proportionately more slowly, leading to an asymptotic relationship between coral cover and fish community metrics (e.g., Holbrook, Schmitt & Brooks, 2008). When coral cover is very high, fish diversity and/or abundance may even start to decline if coral reduces the availability of other microhabitat types (e.g., algal cover) upon which some species depend (Wilson et al., 2009; Glynn et al., 2014).

Our goal was to examine variability in the relationship between live coral cover and reef fish diversity and density to generate insights into the potential responses of fish communities to natural and anthropogenic disturbances on coral reefs. To do so, we replicated a survey of reef fish diversity undertaken 31 years ago by Bell & Galzin (1984) in the lagoon of the Mataiva atoll, French Polynesia. Located far from large land masses, large human populations, and most local human impacts, the reefs of Mataiva, like those of most Polynesian islands, are considered to be under low anthropogenic threat (Salvat et al., 2010; Wilkinson, 2008).

Materials and Methods

Study area

The study was carried out on the atoll of Mataiva, the western-most atoll in the Tuamotu Archipelago, French Polynesia (14°55′S, 148°36′W). This small atoll (10 km × 5 km) has an unusual morphology with a reticulated lagoon divided into approximately 70 pools (average depth: 8 m), separated by a network of slightly submerged coral reef partitions (Delesalle, 1985) (Fig. 1). In February 2012, we reassessed coral cover and fish assemblages in the same 13 pools as those surveyed in 1981 by Bell & Galzin (1984).

Figure 1 The reticulated lagoon of Mataiva, where coral cover and fish diversity and abundance were recorded in 1981 and 2012 (Photo: Suzanne C. Mills).

Estimating live coral cover

Bell & Galzin (1984) estimated live coral cover along a single 50 m transect along the perimeter of each study pool, at depths of 0–3 m. Their estimates of coral cover were qualitative, and they assigned each site to one of five coral cover categories: 0% (all dead), <2%, 2–<5%, 5–10%, and >10%. Bell & Galzin (1984) reported that a subsequent quantitative evaluation of coral cover showed an average of 10.7% (SD = ±9%) for sites in their >10% category, providing support for their qualitative assessment.

In 2012, we estimated the coral cover along three 50 m transects at each site surveyed by Bell & Galzin (1984). Transects were laid parallel to the rim of each pool at an average depth of ≈1.5 m (range: 0.5–3 m), with as much distance as possible between transects. The total length of the reef rim around each pool varied, hence transects were separated by 5–30 m. Using a point-intercept method, we recorded benthic cover type every metre under the transect line (total: 51 points per transect). Benthic cover categories included: live coral (to genus; mainly Acropora, Porites, Montipora and various favid corals), macroalgae (mostly Turbinaria and Halimeda), other benthic organisms (e.g., sponges), rubble, and sand (following Lison de Loma et al., 2008).

To check the accuracy of the point-intercept method, we also took a photograph of the substratum at each intercept point (total: 51 photographs per transect) on 22 of the 39 transects. The shallow depth of some parts of the transects prevented us from standardising the distance between camera and reef, hence the size of the photoquadrats varied between 100 cm2 and 625 cm2. We placed a digital grid on each photoquadrat and estimated percent live coral cover visually. We calculated average coral cover for each transect by weighting the coral cover of each photograph by the area it covered. The percent of coral cover derived from the point-intercept method was significantly correlated with the visual assessment of photoquadrats (r = 0.91, N = 22 transects, P < 0.0001). We therefore used coral cover from the point-intercept survey in subsequent analyses.

Estimating fish diversity and density

To estimate fish diversity and density, we faithfully replicated the method of Bell & Galzin (1984). At each site, along one of the transects laid for coral assessment, we recorded the number of each species of reef fish within 2.5 m on either side of the transect line. Data were collected once by each of two observers at a 5 min interval. We did deviate from Bell & Galzin (1984) by surveying fish on each of the other two transects laid for coral cover assessment. Each observer surveyed one of these two additional transects.

Data analysis

Changes in live coral cover

The cover of each benthic category was expressed as the number of intercept points at which a category was present, divided by the total number of intercept points per transect. To obtain total coral cover comparable to Bell & Galzin (1984), we summed the cover of all live coral genera within each transect. In analyses requiring all three transects surveyed, values were averaged across the transects at each site.

To assess changes in the percent coral cover, we conducted paired comparisons between live coral cover at each site in 1981 (from Bell & Galzin, 1984) and in 2012 (this study). Because Bell & Galzin (1984) reported coral cover in categories, we conservatively assigned the upper bound of each category to each site for 1981. For their highest category (>10%), we used a value of 20% coral cover, which is the mean coral cover + 1 SD obtained in the subsequent quantitative coral survey. The results are similar if we assume that coral cover at the most coral-rich site in 1981 was as high as the highest cover we observed in 2012 (i.e., 45%) so we elected to use only the lower (20%), more conservative cover estimate. We performed one comparison between coral cover in 1981 and that obtained from a single transect (i.e., the transect surveyed by both fish observers) in 2012, which mirrors the sampling effort of Bell & Galzin (1984), and one comparison using the average of the three 2012 transects. The results are similar, hence we present only the former below but show the additional results in the online supplement (Fig. S1A).

Changes in reef fish community composition

We compiled lists of reef fish species observed on the transects at each site in 1981 and 2012 to obtain estimates of fish species richness. We counted the numbers of species recorded in either and both surveys.

There was strong agreement between the two fish observers. The two sets of counts were significantly correlated at each of the 13 sites (mean correlation coefficient r ± SE: 0.88 ± 0.035), and the average slope of correlations across sites (1.1 ± 0.13) did not differ significantly from unity (one-sample t-test, t12 = 0.76, P = 0.46). We thus averaged density estimates across the two observers at each site, as did Bell & Galzin (1984). We compared fish density (numbers per 250 m2) in 1981 and 2012, first using only the jointly surveyed 2012 transect to match the sampling effort of Bell & Galzin (1984), and then using the average of the three 2012 transects. Again, we present only the former here but the additional results can be found in the online supplement (Figs. S1B and S1C).

To examine differences in fish assemblages between years, we ran a permutation-based, non-parametric multivariate analysis of similarity (ANOSIM; Clarke, 1993) using PRIMER (v. 1.0.3; Clarke & Gorley, 2006). Abundance matrices (species by site) were compiled for 1981 and 2012, and the data were square-root-transformed to reduce the influence of very abundant species. Bray-Curtis similarity coefficients were computed between pairs of pools (Clarke & Warwick, 2001). The ANOSIM procedure was carried out on the similarity matrix. ANOSIM generates an R statistic, which varies between 0 (similarities within and between samples are the same) and 1 (all samples within groups are more similar to each other than to any sample across groups) and is tested for difference from zero with a permutation test (in this study, N = 999 permutations). We visualized the differences in fish assemblages with a non-metric multidimensional scaling (MDS) plot in which samples that are more similar in community composition appear closer together than more dissimilar samples. Stress values of <0.1 suggest that distances among samples in an MDS plot accurately reflect the extent of community differences (Clarke & Warwick, 2001). We used the Multivariate Dispersion (MVDISP) procedure to estimate and compare sample dispersion, i.e., variability in assemblages across pools in each year. Finally, we conducted a similarity percentages (SIMPER) analysis to identify the main taxa responsible for the differences observed between 1981 and 2012. We deemed taxa to be important to group differences if their individual dissimilarity contribution was 2.3% or more, which is twice the expected value if dissimilarity were evenly partitioned among all taxa in the analysis (i.e., 100% divided by 87 taxa, multiplied by two). The evenness of each taxon’s dissimilarity contribution across sites was also considered by examining the Consistency Ratio (CR), which is calculated by dividing a species’ average dissimilarity contribution by the standard deviation in dissimilarity values (of that species, for the groups being compared). A CR value >1 indicates a taxon that contributed fairly equally across all samples (Terlizzi et al., 2005).

To examine shifts in reef fish trophic structure, we assigned each positively identified species to a specific trophic group (i.e., herbivore, coralivore, invertivore, piscivore or planktivore) based on the main dietary items reported in Fishbase (Froese & Pauly, 2014).

Association between live coral cover and fish community

We investigated variation in the relationships between live coral cover and fish species richness and density by fitting linear regressions between these variables for 1981 and separately for 2012. Combining the datasets would have violated the assumption of independence of data points since the same sites were sampled 31 years apart. Other assumptions of parametric testing were met. We again assumed that coral cover in Bell and Galzin’s highest coral category was 20%.

Finally, we further examined the association between live coral and fish by comparing the magnitude of changes in density from 1981 to 2012 between fish species that rely heavily on live coral and those that do not. We categorised fish species as reliant on coral if (1) they use specific coral species as shelter or were described as exclusively associated with coral-rich areas, or (2) they commonly consume coral polyps, as stated on Fishbase (Froese & Pauly, 2014; see Table S1). Coral reliance was not assessed for 10 taxa that were not identified to species.

Results

Changes in live coral cover

In 1981, 4 of 13 sites (31%) had no live coral, and maximum coral cover was over 10% at just one site. In 2012, all sites had some coral (Fig. 2A), and the two lowest coral cover categories used by Bell & Galzin (1984) (i.e., 0% and <2%) were no longer represented. Mean coral cover in 1981 was approximately 4% (± 5.6% SD), assuming that coral cover was 20% at the most coral-rich site. Live coral cover was significantly higher in 2012, with a mean of 28.7% (± 14.3%) (paired t-test, t12 = 6.3, P < 0.0001; Fig. 2A). Mean coral cover derived from three transects at each site in 2012 was 24.4% ± 10.9% (Fig. S1A).

Figure 2 Changes in (A) live coral cover, (B) fish species richness (# fish species per 250 m2), and (C) fish density (# fish per 250 m2) on coral reefs of the Mataiva lagoon between 1981 and 2012.

Each line represents a site, and the trajectory for each site is colour-coded to match the initial category of coral cover reported for 1981 in Bell & Galzin (1984): 0%, black; <2%, blue; 2–<5%, green; 5–10%, orange; >10%, red. Some points with similar values are offset for clarity.

Changes in reef fish community

In 1981, Bell & Galzin (1984) recorded 61 fish species across the 13 pools surveyed. In 2012, using the same method (i.e., a single 50-m transect per pool), we found 69 species (Table S1). Forty-three species (49% of all species seen on transects) were recorded in both years; 18 species (21%) were found only in 1981, and 26 species (30%) only in 2012. Eleven (42%) of the species recorded in 2012 but not in 1981 were noted during additional roving dives or rotenone surveys by Bell & Galzin (1984). Therefore, only 15 species were recorded in 2012 but not at all in 1981. Interestingly, those species that we did not record in 2012 were significantly less abundant in 1981 than species that we did record in the recent survey (1981 density of species recorded in 2012: 1.4 ± 3.8 fish per 250 m2, not recorded in 2012: 0.3 ± 0.5 fish per 250 m2; t-test for unequal variances, t75.7 = 2.32, P = 0.02). Conversely, species that were not recorded in 1981 were significantly less abundant in 2012 than species recorded in the early survey (2012 density of species recorded in 1981: 3.0 ± 8.4 fish per 250 m2, not recorded in 1981: 0.5 ± 0.9 fish per 250 m2; t-test for unequal variances, t62.9 = 2.29, P = 0.03)

We recorded significantly more fish species per site in 2012 (25.2 ± 6.7 species 250 m−2) than in 1981 (15.1 ± 8.2 species 250 m−2) (paired t-test, t12 = 4.12, P = 0.001; Fig. 2B). Fish density was also higher in 2012 (193.0 ± 110.2 fish 250 m−2 versus 98.5 ± 77.4 fish 250 m−2 in 1981) (paired t-test, t12 = 3.1, P = 0.009; Fig. 2C). The majority of species (59%, or 51 out of 87) increased in abundance between the two surveys. In 1981, only six taxa (Cheilodipterus quinquelineatus, Chaetodon ephippium, Chromis viridis, Scarus spp., Acanthurus triostegus and Amblyogobius phalaena) were present at 9 or more of the sites (Table S1). In 2012, these six taxa remained widespread, but nine more species (Chaetodon auriga, C. lunulatus, Abudefduf sexfasciatus, Dascyllus aruanus, Halichoeres trimaculatus, Thalassoma hardwicki, T. quinquevittata, Chlorurus sordidus and an unidentified goby) were found at most sites (Table S1).

In general, fish species density estimates in 1981 and 2012 were positively related (r = 0.38, N = 87 species, P < 0.0001), but there were mismatches that were reflected in the community analysis. The fish community composition was significantly different in 1981 and 2012 (ANOSIM, R = 0.40, P = 0.001). The MDS plot revealed little overlap in fish assemblages between 1981 and 2012 (Fig. 3), and assemblages in 1981 were less similar to each other (within-group similarity = 37.3%; dispersion value = 1.24) than in 2012 (within-group similarity = 49.2%; dispersion value = 0.76). Ten taxa in six families contributed disproportionately to differences in community composition between 1981 and 2012 (Table 1). Seven of these species were more abundant in 2012.

Figure 3 Multidimensional scaling plot of reef fish community composition from the Mataiva Atoll lagoon, French Polynesia.

Each point represents the reef fish assemblage of a pool within the lagoon in 1981 (red points) and 2012 (blue points). The low stress value (0.08) suggests that the three-dimensional depiction shown accurately reflects differences among communities.

Table 1 Reef fish taxa that contributed disproportionately to dissimilarity in fish community composition between 1981 and 2012 on coral reefs of the Mataiva lagoon.

Mean densities (# per 250 m2 ± 1 SD), consistency ratios, and individual and cumulative contributions (in %) to differences between years are shown. The consistency ratio is calculated as a species’ average dissimilarity contribution divided by the standard deviation of dissimilarity values. The species in grey has a consistency ratio <1, which indicates an uneven contribution to community dissimilarity across sites. The analysis was conducted on square-root-transformed data (see Methods) but untransformed densities are presented here.

Species	Mean density (SD)	Consistency
ratio	Individual
contribution (%)	Cumulative
contribution (%)	
	1981	2012				
Chlorurus sordidus	9.9	51.1	1.62	10.64	10.6	
	(18.9)	(33.7)				
Chromis viridis *	26.7	34.1	1.37	8.46	19.10	
	(29.3)	(49.3)				
Dascyllus aruanus *	7.2	23.2	1.20	6.08	25.17	
	(11.5)	(29.5)				
Acanthurus triostegus	5.0	2.8	1.30	4.11	29.28	
	(6.9)	(10.4)				
Thalassoma hardwicke	1.2	6.9	1.49	3.77	33.05	
	(2.0)	(6.1)				
Stegastes nigricans *	1.5	10.7	0.64	3.51	36.56	
	(4.2)	(21.4)				
Scarus spp	8.6	5.5	1.25	3.05	39.61	
	(14.9)	(5.0)				
Chaetodon lunulatus *	2.3	4.3	1.45	3.04	42.66	
	(4.8)	(4.4)				
Halichoeres trimaculatus	0	3.9	1.24	2.87	45.53	
	(0)	(5.1)				
Amblygobius phalaena	10.2	8.7	1.26	2.42	47.94	
	(9.0)	(5.6)				
Notes.

* species considered to be reliant on live coral for shelter or food.

Reef fish trophic composition shifted between the two surveys. While the proportion of herbivores and invertivores remained constant (23% and 46%, respectively) across years, there was a 2.5-fold decline in the proportions of coralivores and piscivores (from 16% to 6% in both cases). Planktivores were absent in 1981 but accounted for 18% of species in 2012 (Table S2).

Changes in the relationship between live coral cover and reef fish community

In 1981 there were significant, positive relationships between live coral cover and fish species richness (R2 = 0.83, F1,11 = 51.7, P < 0.0001) as well as fish density (R2 = 0.65, F1,11 = 20.2, P = 0.001). These relationships were no longer found in 2012 (fish species richness: R2 = 0, F1,11 = 0.002, P = 0.97; density: R2 = 0, F1,11 = 0.003, P = 0.96) (Fig. 4).

Figure 4 Relationships between live coral cover and (A) reef fish species richness (# fish species per 250 m2) and (B) density (# fish per 250 m2) in 1981 (red points and regression lines) and in 2012 (blue points and regression lines).

Superimposed points are offset slightly for clarity. Regression equations (A) 1981: Y = 0.53∗X + 11.9; 2012: Y = −0.006∗X + 25.4; (B) 1981: Y = 11.1∗X + 53.1; 2012: Y = −0.12∗X + 196.4.

Nearly three-quarters (73%) of species that rely on live coral for shelter or food increased in density between 1981 and 2012, compared to 57% of species that are not coral-reliant. This association was not significant (X12=1.52, P = 0.22). In addition, there was no difference in the magnitude of density change from 1981 to 2012 between species that do and do not rely on live coral (change in density of coral-reliant species: +1.65 ± 3.7 fish per 250 m2; non-reliant species: +1.05 ± 5.7 fish per 250 m2; t75 = 0.47, p = 0.64).

Discussion

Coral cover and reef fish communities in the Mataiva Lagoon changed markedly between 1981 and 2012. Live coral cover increased 6-7 fold and fish density nearly doubled. While overall reef fish richness in the lagoon did not increase appreciably, there was a significant shift in the fish community composition. The strong positive relationships between live coral cover and fish richness and abundance noted in 1981 when coral cover rarely exceeded 10% (Bell & Galzin, 1984) were no longer present in 2012 when coral cover did not fall below this value. The most parsimonious explanation for these contrasting relationships is that, over the combined range of coral cover observed in 1981 and 2012 snapshots, there is an asymptotic relationship between reef fish and corals. Our results, and other data from the south and west Pacific, suggest that fish diversity and abundance might accumulate rapidly up to a threshold of approximately 10% live coral cover. Such a relationship would have implications for our expectations of resistance and recovery of coral reefs to major disturbances.

In contrast to the general patterns of decreasing coral cover reported for many parts of the Indo-Pacific region (e.g., Bruno & Selig, 2007), there was a marked increase in coral cover at the majority of sites in the Mataiva lagoon. The exact trajectory of change and the reasons for this improvement in coral abundance are not clear. Anthropogenic impacts on Mataiva reefs are relatively limited because of the small human population (<300 people), the scarcity of pollution sources, and the low fishing pressure (R Beldade, pers. obs., 2012). Coral reefs in the region have experienced multiple natural disturbances, but records for Mataiva—especially for the lagoon—are limited. Bell & Galzin (1984) surmised that the average coral cover was very low (4%) in 1981 because of coral mortality caused by prolonged low tides in the lagoon six months prior to their survey. Thereafter, the atoll was affected by two successive cyclones in 1983, an algal bloom in 1988, and mass coral bleaching in 1998 (Adjeroud et al., 2005). The abrupt drop in coral cover at a site monitored irregularly on the outer slope of the atoll between 1994 (25%) and 1999 (5%) was attributed mainly to coral bleaching in 1998 (Adjeroud et al., 2005). Corals within the Mataiva lagoon may have also been affected by bleaching, similarly to corals in the Rangiroa lagoon, 80 km away, which experienced severe mortality during the same warming event (Mumby et al., 2001). Later cyclones and bleaching events that have not been reported might have also had impacts on Mataiva corals. The recovery of the Mataiva corals, from 4 to 30% cover, is therefore likely to have been uneven over time, and has probably occurred over a shorter period (i.e., since the last unrecorded disturbance) than the 31-year span of our two snapshots. Such a potentially rapid, post-disturbance increase in coral cover is not unprecedented (e.g., Diaz-Pulido et al., 2009; Gilmour et al., 2013; Roff et al., 2014), but we cannot determine whether recovery indicates resilience, i.e., a return to pre-disturbance community structure (Connell & Sousa, 1983), because of the lack of early, species-specific coral cover data.

Reef fish communities can show a range of responses to coral recovery. Some communities fail to return to pre-disturbance levels of species diversity and abundance, even after long periods of time (e.g., Bellwood et al., 2012). Others appear to be truly resilient and, eventually, have community structures that are almost indistinguishable from the initial assemblages (e.g., Sano, Shimizu & Nose, 1984; Sano, 2000). However, some reef fish communities show recovery of overall diversity and abundance without community resilience (e.g., Halford et al., 2004; Berumen & Pratchett, 2006). It is difficult to assign Mataiva to one of these three patterns without knowing whether the 1981 fish communities represent pre- or post-disturbance assemblages. However, we did find that fish density nearly doubled as coral cover increased but the species composition of the later fish assemblage was significantly different from that of 1981. While all species that were found at the majority of sites in 1981 were found again at the same sites in 2012, several more species became widespread in 2012. Species densities were correlated between the two years, but the relationship was weak, explaining only 14% of variance in density. The majority (60%) of fish species increased in abundance but 40% declined or their abundance remained unchanged. Those that did not increase included species spanning 16 families and a range of body sizes, trophic groups and reliance on coral (Table S1). Ultimately, there was limited overlap in community composition across the years and fish assemblages in 2012 were more homogeneous in composition across sites than they were in 1981 (see also Emslie et al., 2008). This convergence would be expected if there is a non-linear relationship between fish and coral cover (see below).

Interestingly, the total fish species richness in the lagoon remained remarkably similar despite marked increases in site-specific fish species richness and density over time. However, only half of the fish species were reported in both years. The fact that species not recorded in one survey tended to always occur at very low density in the other survey suggests an issue with the detection of rare species, but there might also be real additions or losses of species. Thus, the relatively constant overall species richness over time might belie a temporal substitution of several of the least abundant species (Cheal et al., 2008; Dornelas et al., 2014). Metrics such as total species number and coral cover that mask community composition are therefore poor indicators of community resilience.

Changes in fish species density in Mataiva were not as strongly associated with the reliance of species on live coral as initially expected. In general, fish that depend heavily on live coral for either food (e.g., many butterflyfishes) or shelter (e.g., small damselfishes in branching Acropora) are expected to track closely the abundance of live coral (Jones et al., 2004; Pratchett et al., 2008). Indeed, nearly three-quarters of the species we identified as coral-dependent did become more abundant between 1981 and 2012. It is likely, for example, that a greater availability of branching corals explains the marked increase in relative importance to species richness of planktivores, many of which are closely tied to this habitat type. Coral composition was not recorded in 1981, hence we cannot test this idea. However, more than half of the fish species that do not rely on coral also increased in density. The density of coral-dependent species did not increase more than that of other fish species. Moreover, coral-dependent species at Mataiva did not contribute disproportionately to differences between the 1981 and 2012 assemblages (i.e., 4 coral-dependent species out of 9 species in Table 1 versus 26 out of 77 species overall), and coralivorous species accounted for a smaller proportion of the fish richness in 2012 than in 1981. The recovery of corals must therefore have improved reef habitat in other ways than by simply increasing the availability of live coral. One such way is through increased topographic complexity, which tends to be higher in coral-rich areas and is a recognised determinant of reef fish community structure (Garpe et al., 2006). Many of the fish species that became more abundant in 2012 but were not strictly dependent on live coral per se might depend instead on some of the diverse microhabitats and shelters provided by architecturally complex corals (e.g., Sano, Shimizu & Nose, 1984) . Others might have responded to changes in fish prey availability, driven by benthic habitat change (e.g., Graham et al., 2007).

The relationships between live coral cover and reef fish diversity and abundance were variable over time. Fish richness and density were strongly and positively associated with live coral cover in 1981 when coral cover was very low (<∼10%). This was not the case in 2012, when coral cover varied between 10 and ∼50%. When combined, our two snapshots—taken 31 years apart—suggest a rapidly asymptotic relationship between coral and fish (see also Holbrook, Schmitt & Brooks, 2008; Wilson et al., 2009). Interestingly, the threshold coral cover we infer from this hypothesised relationship for lagoonal reefs at Mataiva (∼10%) is very similar to that noted in a study using space-for-time substitution of natural and experimental reefs in Moorea, French Polynesia (Holbrook, Schmitt & Brooks, 2008) and another using time-series of natural reefs on the Great Barrier Reef (Wilson et al., 2009). This threshold coral cover may therefore be a general feature, at least for coral reefs of the south and west Pacific.

If the hypothesised non-linear relationship between coral and fish is indeed accurate, it has implications for our expectations of resistance and recovery of fish communities on disturbed reefs. It suggests that fish assemblages on most reefs should be able to weather heavy losses of live coral cover without showing concomitantly large responses by fishes, at least in terms of overall diversity and abundance, until live coral cover drops below approximately 10%. Further declines in coral cover should lead to precipitous losses in reef fish. Conversely, the recovery of these simple metrics of fish communities should be swift as coral recovers to the threshold level, although true resilience in the sense of returning to original community structure may take longer, if it occurs at all. A key question to address now is whether the tipping point in coral cover we and others have identified for reef fish communities is affected by the increasing severity and frequency of multiple disturbances on coral reefs.

Supplemental Information

Table S1 Reef fish species, and their abundances (number of individuals per 250 m2), recorded at 13 sites in the lagoon of Mataiva Atoll, French Polynesia

The numerator shows abundance in 1981, as reported by Bell & Galzin (1984); the denominator shows abundance in 2012, derived from the same surveying method (i.e., one 50-m transect) at the same sites. Zeros are omitted for clarity. Asterisks denote species considered to be reliant on live coral for shelter or food.

Click here for additional data file.

Table S2 Proportion of reef fish species which were present in the Mataiva lagoon in1981 but not in 2012, and vice versa, in relation to trophic group

Click here for additional data file.

Figure S1 Change in (A) coral cover, (B) fish richness (# fish species per 250 m2), and (C) fish density (# fish per 250 m2) on coral reefs of the Mataiva lagoon between 1981 and 2012.

The 2012 data represent the mean of three 50-m transects. Each line represents a site, and each site and its trajectory are colour-coded to match the initial category of coral cover reported for 1981 in Bell & Galzin (1984): 0%, black; <2%, blue; 2–<5%, green; 5–10%, orange; >10%, red. Some points with similar values are offset for clarity. Mean coral cover (paired t-test, t12 = 8.4, P < 0.0001), fish richness (paired t-test, t12 = 2.3, P = 0.04) and fish density (paired t-test, t12 = 3.9, P = 0.002) were all significantly higher in 2012 than in 1981.

Click here for additional data file.

Figure S2 Relationships between live coral cover and (A) reef fish species richness (# fish species per 250 m2) and (B) density (# fish per 250 m2) in 1981 (red points and regression lines) and in 2012 (blue points and regression lines).

Superimposed points are offset slightly for clarity. Data for 2012 are derived from the averages of three transects at each site. Regression equations and statistics (a) 1981: Y = 0.53∗X + 11.9; r2 = 0.60, F1,11 = 16.5, p = 0.0022012: Y = −0.036∗X + 21.6; r2 = 0.005, F1,11 = 0.05, p = 0.82 (b) 1981: Y = 11.1∗X + 53.1; r2 = 0.65, F1,11 = 20.2, p = 0.001 2012: Y = −1.3∗X + 242.2; r2 = 0.02, F1,11 = 0.22, p = 0.65.

Click here for additional data file.

We thank Riccardo Rodolfo-Metalpa and Luiza Paoliello for field and data analysis assistance respectively, and the Pension Ariiheevai (especially Francois Tetuira) for their wonderful hospitality and providing the facilities to conduct this study.

Additional Information and Declarations

Competing Interests

Author Contributions

The authors declare there are no competing interests.

Ricardo Beldade conceived and designed the experiments, performed the experiments, analyzed the data, wrote the paper, prepared figures and/or tables, reviewed drafts of the paper.

Suzanne C. Mills conceived and designed the experiments, performed the experiments, contributed reagents/materials/analysis tools, wrote the paper, reviewed drafts of the paper.

Joachim Claudet conceived and designed the experiments, performed the experiments, wrote the paper, reviewed drafts of the paper.

Isabelle M. Côté conceived and designed the experiments, performed the experiments, analyzed the data, contributed reagents/materials/analysis tools, wrote the paper, prepared figures and/or tables, reviewed drafts of the paper.

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
