# Peer review of "More coral, more fish? Contrasting snapshots from a remote Pacific atoll"

_PeerJ, doi:10.7717/peerj.745_

## Round 0.1 · original submission · Minor Revisions

In general, both myself and the reviewers agree that the study is well executed and the interpretation is valid. One point of clarification that both reviewers have requested is why you have selected the 20% threshold, so please make sure you address this comment. The other comments of the reviewers are all straight forward so I am confident you will have no problems addressing those.

Reviewer 1 ·

Basic reporting

No comments

Experimental design

Overall the experimental design and its description were good, but there are a number of clarifications needed:

Line 109 - It was not clear why you chose 20% for the upper limit based on the description you provided from lines 67-72 of Bell & Galzin's data on live coral cover. More clarity is needed for the reader to follow your choice here.

Line 156 need 'for' prior to 2012.

Line 154-157 - You have not provided any explanation of why you have run different analyses for the two years, rather than running a single analysis here, with year:coral cover as an interaction term. I presume it is linked to the lack of overlap in coral cover values for the two years, but currently any explanation is lacking.

Line 166 - 167 - once again your description of 'coral cover was probably...' doesn't seem to match up with what you wrote earlier in the methods regarding the data from 1984. More clarity is needed to describe the coral cover data from 1984.

Validity of the findings

Overall I found the discussion thoughtful and well presented. I have one or two minor comments:

Line 227 - Here (and in the abstract) you state that there is a threshold at approximately 10% - I feel it might be better to put in a wider range of possible values for the threshold based on your description of the 1984 data in the methods/results. Although, this may be because your description of the 1984 coral cover data wasn't 100% clear in the methods.

Line 245-246 - As it stands, sentence is unclear. Need to move 'that have not been reported' to earlier in the sentence, after 'bleaching events'.

Line 249 - Add 'potentially after 'such a', so reads 'such a potentially rapid post-disturbance...', since this rapid increase is inferred rather than having been observed.

Line 263 - Wording of 'remained so in 2012' is unclear.

Line 278 - What are you referring to when you state 'are therefore poor indicators of resilience'? Are you referring to resilience of the community or certain taxa? As presumably the community as a whole can be resilient even if certain taxa are not?

Line 292-295 - Need references here to support your statements.

Additional comments

Overall I found this to be an interesting and well presented study.

·

Basic reporting

This paper is clearly written and adheres to PeerJ guidelines.

Experimental design

The manuscript attempts to compare the effects of changes in benthic composition to observed shifts in fish assemblages. Surveys conducted by the authors in 2012 are compared to the findings of a previous study conducted in 1984. Consequently, the authors were required to simulate the methods of 1984 surveys as closely as possible to obtain comparable results. They have made an admirable attempt to do this, particularly given the earlier paper did not report on some aspects of the benthic communities (i.e. coral species/morphotype etc) that would have provided greater insight into the reason for the observed shifts. However, one point requires clarification:
Bell & Galzin (1984) present coral cover values at each site in categories, rather than absolute values. The authors state that the mean coral cover for sites in the category '>10%' was 10.7%. However, they assume that sites in that category supported coral cover of 20%. Given the mean value for that >10% category was provided in Bell & Galzin, why did the authors assume a different value?

Validity of the findings

The findings of this study are valid, given their results. However, it would be interesting to see a little more interrogation of some aspects of their data set.
Specifically, I would recommend the authors look at other functional groups in addition to coralliavores. Different functional groups respond differently to changes in live coral cover, and examination of other functional groups may provide greater insight into the likely changes in benthos.
It is unfortunate that Bell & Galzin apparently did not record any information on the types of corals present in 1984. However, Berumen and Pratchett (cited in the Introduction) reported significant shifts in coral and fish assemblages in French Polynesia, despite no long-term changes in coral cover. Examining changes in other functional groups (particularly herbivores) may provide some clues regarding a possible shift in benthic community composition influencing the fish assemblage.

Additional comments

Along the same lines as the point above, it would be interesting for the authors to look at which fish species disappeared between 1981 and 2012, and which appeared. I note that the authors attribute some turnover of species to rarity and detection issues, which is an important consideration. However, apart from rarity, was there a particular functional group disproportionately represented in the turnover between 1984 and 2012?


Minor comments:
Line 159 - 2012, not 2912

Line 243: Bleaching mortality in Rangiroa does not necessarily imply bleaching occurred in Mataiva. Bleaching is often heterogeneous over much smaller spatial scales than 80 km, and I would be reluctant to assume that Mataiva bleached on the basis of bleaching being observed 80 km away.

Line 266: 60% od species increased in abundance but 40% did not. Did these species decline or remain stable? Again, analysis of the type of species that declined (if any patterns are evident) would be helpful. If there is no pattern, at least state that you have looked into it and no pattern was evident.

Line 277: 'Constant species richness belies temporal substitution of the least abundant species' - this could also be a detection issue. Looking at functional groups will add more weight to whether this is an actual shift or not.

---

## Round 0.2 · Minor Revisions

The manuscript has improved somewhat based on the Reviewer comments but there are still a few minor issues to address. Number 1 is essential. All comments are easy to address and I look forward to seeing the manuscript with these small changes soon.

1. In the Discussion, you refer repeatedly to a non-linear relationship between coral cover and fish species richness/density as one of your main results. However, you have only used linear relationships in your regression. At the moment, you cannot say that there is greater support for an asymptotic relationship compared with a linear relationship because you have not tested it (although it certainly looks that way from the plots). It would be easy for you to do this by adding a quadratic term to the predictor variable (i.e., coral cover) in the regression, and then comparing the adjusted R-squared and/or the AICc, which should increase/decrease respectively. Please do this to strengthen your case for an asymptotic relationship in the data.

2. I agree with the point of Reviewer 2 regarding breaking down your results to fish functional and/or trophic groups of fish. You show an interesting shift in your analysis within the response to comments and, despite having no clear explanation for it given the data available, it is still an interesting pattern to show in the manuscript. Please add these data to the manuscript as Reviewer 2 suggested and try to offer a hypothesis about the possible causes.

3. Please change the word “variable” in line 3 of the abstract. It is a bit confusing because it sounds like a statistical term. Individualistic? Idiosyncratic?

4. Please re-phrase the sentence on L110-111 “When needed, values were then averaged across the three transects at each site”. What does “when needed” mean?

5. L120. There seems to be a word missing in this sentence around “…that obtained…”

6. L127. Please change “tallied” to “summed”

7. L284. I think you have an extra “remained” here

---

## Round 0.3 · Minor Revisions

I am happy with all of the changes apart from the asymptotic regression models, which you must address more thoroughly before I will accept this manuscript for publication.

To reiterate - this is claimed as your MAIN result but right now is backed up poorly. Therefore, you must change your METHODS and RESULTS (main text, tables, figures) to reflect this, not simply add a supplementary table and a sentence in the text.

Methods
- explain that you used non-linear regression and justify why. It is not enough to put this in the legend for Table S3.
- e.g., is the asymptotic relationship achieved by adding a quadratic term to the model?
- explain how you compare the linear vs non-linear models.

Results
- Figure 4 should show the asymptotic relationship, NOT the linear relationship because you say the latter is not very good - so why would you show a figure of it instead of the better non-linear fit?
- Also for Fig.4, if you have a non-linear fit, this should be part of any regression equation. If you do not know how to do this, a useful book is titled "How to be a Quantitative Ecologist". Check it out.
- Table S3 should be included in main text. Label "a" and "b" by their proper statistical terms (e.g., intercept and coefficient).
- It is unclear whether your coefficients are reported in the text are from the linear or non-linear model. It should be the latter.
- if you are unsure how to present non-linear regression results, have a look at other papers that have done this and try to emulate the style.

---

## Round 0.4 · Minor Revisions

I understand your objections and agree this is not possible for all your data given the non-independence. I think the root of the confusion lies in the title, abstract and discussion where the asymptotic, particularly "functional", relationship is mentioned. I have been through the manuscript and identified the lines of text that would benefit from being modified to make it much more clear that the relationship is a hypothesis and not testable with the data you have. It is a subtle difference but is important because, as you rightly point out in your discussion, there are other things that differ between these time slices that may have influenced the relationship. Please decide whether you wish to include Table S3 or not given this discussion.

Title.
Last 3 lines of abstract - the first of these is perfect, the next 2 lead to confusion.
L172. Clarify why these are fit separately i.e., non-independence of data points
L231. Please add in here that the diagnostics were ok e.g., residuals were normally distributed. People may be wondering why a Poisson error distribution (commonly used for count data) was not necessary.
L250-254.
L296-297.
L333 --> this is great, don't change.
L343. You need to be careful with this final paragraph and make clear the asymptotic relationship is hypothesised based on your results and is not a result per se.

These all require only small modifications to the text so I look forward to receiving the final manuscript.

---

## Round 0.5 · accepted · Accept

I am happy to accept your manuscript for publication. Congratulations.